# Recent Implementations of Hydrogel-Based Microbial Electrochemical Technologies (METs) in Sensing Applications

**DOI:** 10.3390/s23020641

**Published:** 2023-01-06

**Authors:** Zeena Wang, Dunzhu Li, Yunhong Shi, Yifan Sun, Saviour I. Okeke, Luming Yang, Wen Zhang, Zihan Zhang, Yanqi Shi, Liwen Xiao

**Affiliations:** 1Department of Civil, Structural and Environmental Engineering, Trinity College Dublin, D02 PN40 Dublin, Ireland; 2TrinityHaus, Trinity College Dublin, D02 PN40 Dublin, Ireland

**Keywords:** microbial electrochemical technology, microbial sensors, microbial fuel cells, hydrogels, biosensors, microbial electrolysis cells

## Abstract

Hydrogel materials have been used extensively in microbial electrochemical technology (MET) and sensor development due to their high biocompatibility and low toxicity. With an increasing demand for sensors across different sectors, it is crucial to understand the current state within the sectors of hydrogel METs and sensors. Surprisingly, a systematic review examining the application of hydrogel-based METs to sensor technologies has not yet been conducted. This review aimed to identify the current research progress surrounding the incorporation of hydrogels within METs and sensors development, with a specific focus on microbial fuel cells (MFCs) and microbial electrolysis cells (MECs). The manufacturing process/cost, operational performance, analysis accuracy and stability of typical hydrogel materials in METs and sensors were summarised and analysed. The current challenges facing the technology as well as potential direction for future research were also discussed. This review will substantially promote the understanding of hydrogel materials used in METs and benefit the development of electrochemical biosensors using hydrogel-based METs.

## 1. Introduction

Microbial electrochemical technology (MET) is a fast-expanding field of research that utilises the metabolism of electrogenic microbes to catalyse oxidation and reduction reactions that occur in the anode and cathode [1,2]. The electrogenic microorganisms are able to release electrons through a variety of electron transfer methods and the electron is then passed from the anode to the cathode to produce an electrical output. MET is an overlap of a variety of research areas including microbiology and electrochemistry, material sciences, environmental and electrical engineering, etc. [3,4]. Its ability to sustainably utilise and convert a wide range of products in any form (solid, liquid, or gas) into useful products such as electricity and biofuels makes it highly applicable in a wide variety of research fields [5].

Microbial fuel cells (MFCs) and microbial electrolysis cells (MECs) are the most typical METs widely used for wastewater treatment and green energy production. In recent literature, they have garnered significant interest in the sensor industry due to their versatile biosensing properties and ability to use microorganisms as a biocatalyst [6]. They have been developed as the sensor detecting organic matters (e.g., biological oxygen demand—BOD [7]), nutrients (e.g., nitrate ions [8]), toxicants (e.g., heavy metals [9,10]) in water, wastewater, soil, and human fluid (see Figure 1). Additionally, the ability of MFCs and MECs to act as energy sources demonstrates the potential of these technologies to act as alternative green power supplies for sensors [11].

Hydrogels have great advantages when applied to METs and sensor technology development. For instance, a typical biosensor consists of four parts: the analyte, the analyte binding substrate, the transducer, and the data processor (shown in Figure 2A) [12]. The stability and sensitivity of a biosensor are crucial during its application. Hydrogels are composed of a network of three-dimensional crosslinked polymers that are able to absorb large amounts of liquid [13]. They have a range of unique properties including swelling behaviour, biocompatibility, non-toxicity, porous structure, and self-healing [14], which makes them extremely versatile and suitable for improving the stability and sensitivity of the sensor. Hydrogel networks can either be chemically or physically crosslinked, ensuring structural stability during water absorption [15]. This allows the hydrogel to immobilise the biological substrate as well as create a microenvironment in which the analyte can be confined, thus improving the sensitivity [16]. These properties make hydrogels extremely suitable for electrochemical biosensing applications and development [17].

The high versatility and unique properties of hydrogels have led to significant interest in hydrogel applications in MET and biosensors. Key terms search within the web of science databases shows that over 4700 studies have been conducted on hydrogel research within the sensors and biosensors sector (Table 1). It has become increasingly popular to employ hydrogel in the development of METs, with 70+ publications. With a greater demand for sensors with higher sensitivity and response times, it is important to understand the current advancements happening within the sectors of hydrogel METs and sensors. Understanding the benefits and drawbacks of typical hydrogel-based METs and their applications in biosensors will substantially promote future development and application of this technology.

Here, we aim to give an overview of the applications of hydrogels within the sensor and MET sector, with a specific focus on MFC and MEC technology. The current literature on hydrogel applications in MFC-based sensors will be summarised and the limitations, barriers, and future trends will be discussed. This review will substantially promote the understanding of hydrogel materials used in METs and benefit the development of electrochemical biosensors using hydrogel-based METs.

## 2. Microbial Fuel Cell and Microbial Electrolysis Cell

MFCs and MECs are two widely researched METs [18]. Although the technical setup of the two types of METs is similar, their reactions, process, and products differ (Figure 1).

A typical MFC contains an anodic and cathodic chamber separated by a proton exchange membrane (PEM) shown in Figure 1A [19]. An electrogenic biofilm containing anaerobic bacteria is able to grow on the anode and produce electrons and protons through the oxidation of substrates [20]. The electrons are transported via an external circuit to the cathode where it reacts with the oxygen and protons to produce water. The flow of electrons is able to generate an electrical voltage [5].

MEC is a modified version of the MFC technology whereby the protons and electrons generated by the microbial biofilm are used to produce biofuel materials such as hydrogen gas and methane (Figure 1B) [19]. Unlike the MFC, the cathode chamber of the MEC is placed under anaerobic conditions. This means that electrons transferred from the anode to the cathode are used to reduce water and protons to produce hydrogen gas [21]. This reaction is not thermodynamically favourable meaning that an additional 0.2–0.8 V needs to be supplied to aid the electron transfer [22,23]. The carbon dioxide produced at the anode can also react with protons and electrons to produce methane rather than be released like in the MFC [19].

The microbial community and microbe-electrode interaction play crucial roles in the efficiency of METs [18]. Currently, many METs are highly limited by poor long-term stability due to the decreased electroactivity over time [24], which is a result of the harsh and fluctuating environmental conditions. Hydrogels could be a solution for this challenge [18]. The porous structure of the gel allows space for microbial colonisation whilst the unique swelling properties maintain a stable internal environment ensuring the long-term viability of the microbes [25]. The gel also allows for easy diffusion of nutrients and waste in and out of the matrix, providing the microbes with the necessary materials for survival [26]. The ability to modify the functional group of matrix polymers can also assist with microbe binding and electron transfer [27]. A study by Ming et al. encapsulated a wound-healing probiotic into a hyaluronic acid hydrogel to protect it from immune system attacks [28]. Du et al. demonstrated the protective ability of hydrogels by subjecting hydrogel-immobilised bacteria to acid attacks [29]. The immobilised bacteria exhibited excellent electrocatalytic activity with an increase in current density.

## 3. Hydrogels Used in METs

### 3.1. Hydrogels for Microbial Immobilisation

In the last ten years, hydrogels have become a common material used for microbial immobilisation, which is crucial for METs [15]. The highly porous structure and rigid matrix provide a stable environment, protecting microbes from environmental fluctuations shown in Figure 2B [30]. This benign environment can suppress outside noise, thereby improving sensing signals [31]. Singh et al. immobilised five strains of strontium-resistant bacteria into an acrylamide hydrogel polymer and found that the immobilised microbes had a higher rate of strontium removal efficiency [32].

Multiple researchers have used alginate [33], cellulose [34], gelatine [35], and silica [36] hydrogel to immobilise different bacterial species for a variety of sensing applications. *E. coli* is commonly used as the model organism in many studies due to its fast growth and easy manipulation. Other popular bacterial species used include *S. oneidensis* [37], *G. oxidans* [38], and *Lactobacillus* sp. [39]. Microbes often have to be cultured just before use as they have been shown to lose their biological activity when placed in storage [40]. Immobilising *E. coli* in poly(vinyl alcohol) (PVA) hydrogels showed exceptional biological activity even after 40 days of storage [41].

Electrochemical sensors often employ electrogenic microbes for sensing as the microbial metabolism can convert chemical energy to electrical energy, forgoing the need for transducers [42]. The electrogenic microbes are often found in consortiums in the form of stable biofilms; however, this can decrease the selectivity of the species. Single species are beneficial as organic consumption is directly linked to the voltage output; however, single-species biofilm is rarely seen in nature and tends to be unstable [43]. Hydrogels can act as an artificial matrix, allowing for single species to be embedded. Kaiser et al. embedded *S. oneidensis* into a PVA hydrogel anode and compared the electrochemical performance to an anode only containing a natural biofilm [44]. The hydrogel-embedded anode showed an improved voltage output. For highly electrogenic pure culture microbes that lack the genes for biofilm formation, hydrogels can also improve immobilisation [27]. Evidently, hydrogel shows great advantages to carry and immobilise bacteria and benefit their performance.

### 3.2. Hydrogel-Based MFCs

Hydrogels have been tailored and adapted for applications in MFC research to improve the metabolic activity of the anodic biofilm and electron transfer efficiency. Hydrogel-based anodes, cathodes, separators/membranes, and electrolytes have been reported in multiple studies. Figure 3 demonstrates examples of each while Table 2 summarises the details.

#### 3.2.1. Anode Hydrogels

##### Conducting Polymer Hydrogels

Conducting polymer hydrogels (CPHs) are a class of materials that combine the high electrocatalytic activity of conductive polymers, with the porous structure of the hydrogel [45]. They are used in the anodes of MFCs as they can physically interact with cell membranes, aiding in the facilitation of electron transference [46]. The improved electrocatalytic activity reduces the electron transfer resistance, and the hydrogel encapsulates microbes in a buffered environment, thereby promoting metabolic activity [47].

Throughout literature, the fabrication of CPHs is commonly observed for anode fabrication [48,49,50]. Polyaniline (PANI) and polypyrrole (PPy) have often been used to modify anodes due to their good electrical conductivity and bioadhesive properties [46]. Cellulose hydrogels are often used in conjunction with PANI and PPy. In parallel studies, Mashkour et al. tested the power density of PANI-Bacterial cellulose (BC) and PPy-BC anode against a graphite plate anode [51]. The PANI-BC produced a maximum power density of 117.76 mW/m^2^ whereas PPy-BC yielded a slightly higher power density of 136 mW/m^2^. Further conductivity improvement was demonstrated by incorporating titanium dioxide into the PANI-BC construct [52].

The addition of multiple conductive materials into hydrogels has demonstrated an improvement in electrical conductivity. Szöllősi et al. created a composite hydrogel containing three electrically conductive materials (Alginate-PANI-titanium dioxide-graphite composite hydrogel) shown in Figure 3A [53]. Although the addition of 0.05 g/mL of PANI and graphite separately yielded a ten-fold increase in conductivity, the addition of PANI and graphite together enhanced the conductivity 105-fold. However, increasing the carbon materials results in the collapse of the gel matrix. A compromise concentration of 0.02 g/mL and 0.05 g/mL of PANI and graphite respectively demonstrated improved conductivity and was able to run continuously for 7 days. Wang et al. created an alginate, PANI, and carbon brush (CB) electrode to monitor chemical oxygen demand (COD) removal [54]. The MFC resulted in a power output of 515 mW/m^2^, 1.5 times greater than the bare anode. Wang et al. further constructed PPy, carboxymethyl cellulose (CMC), and carbon nanotubes (CNT) construct on a CB. The power density output of 2970 mW/m^2^ was 4.34 times greater than the bare anode. This large difference in comparison to the PANI study would mainly be due to the addition of CNTs [55].

Mixing two types of conductive polymers has been shown to also increase the power density. PANI and PPy hydrogel anodes have been shown to have similar power densities (2737.12 and 2859.53 mW/m^3^); however, the PANI-PPy hybrid hydrogel anode had a noticeably higher power density (4413.03 mW/m^3^). Further addition of CNT and Fe_3_O_4_ into the composite reduced internal resistance improving power density [56].

**Figure 3 sensors-23-00641-f003:**
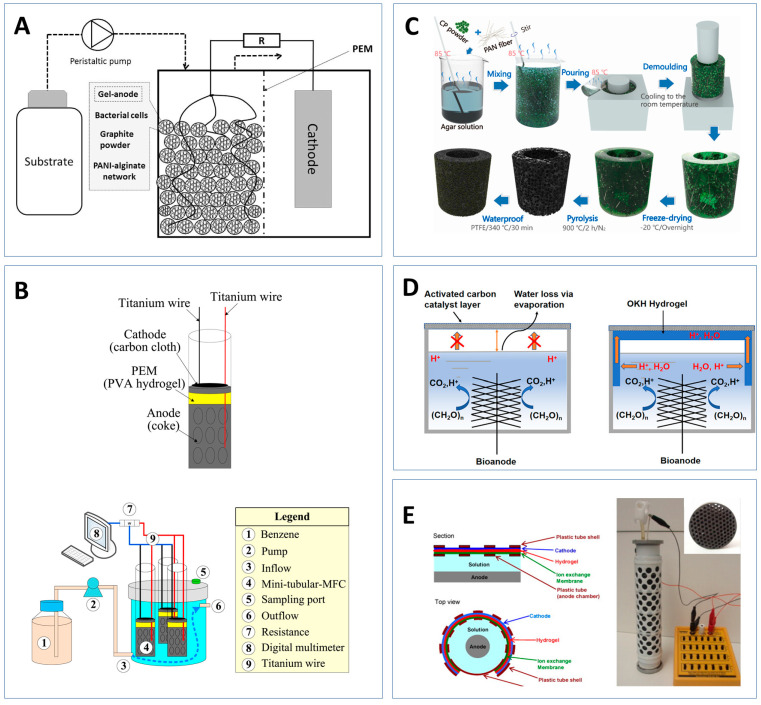
Example of hydrogels applied to different MFC components. (**A**) An alginate/PANI/titanium-dioxide/graphite hydrogel composite with immobilised bacterial cells as an bioanode [53]; (**B**) a tubular MFC with a polyvinyl hydrogel separate membrane used for benzene removal [57]; (**C**) a free-standing cathode produced from polyacrylonitrile fibre, agar, and *Chlorella pyrenoidosa* [58]; (**D**) an oxidised konjac glucomannan/2-hydroxypropytrimethyl ammonium chloride chitosan (OKH) hydrogel was used for ion bridging for an horizonal cathode MFC [59]; (**E**) an ECG gel used as an electrolyte to improve membrane–cathode contact area in a tubular MFC [60].

##### Carbon Composite Hydrogels

CNTs are a class of nanostructures that are increasingly being applied to the anode of MFC due to their ability to improve the electrocatalytic activity of microbes [61]. In 2014, Lui et al. fabricated a CNT-chitosan hydrogel anode by electrodepositing CNT–Chitosan onto a carbon paper anode [24]. The CNT was able to improve the power and current density by allowing direct electron transference between the cytochrome enzyme and the anode. Further improvement of MFC performance can be achieved by incorporating CNT into a CPH. A graphene oxide (GO), CNT, and Poly N-Isopropylacrylamide (PNIPAM) hydrogel anode showed a 100% increase in power and current densities [62]. PPy is another common conductive polymer commonly forming a composite with CNT. Examples in the literature include the production of a PPy-CNT hydrogel [61], and PPy, CMC and nitrogen-doped CNT hydrogel [63], both showing improved power output.

GO is another common material used within electrodes due to its hydrophilic nature and large surface area, supporting microbial adhesion and growth. GO is not a conductive material; however, it can be reduced by some species of microbes and act as a conductor [64]. According to a study by Yoshida et al., the growth of certain electrogenic microbial species has been linked to GO reduction, and the resulting structure is an reduced graphene oxide (rGO) microbial complex [65]. This rGO hydrogel complex has since been used for electricity recovery from dialysis wastewater using MFCs, however, the current output was lower than expected [66]. This is explained by incomplete GO reduction, therefore longer incubation between GO and the microbial species is required [67].

#### 3.2.2. Cathode Hydrogels

Although a large body of hydrogel-based MFC research is dedicated to improving the microbe–anode interactions and electron transfer efficiency, some studies have applied hydrogels to the cathode of MFC to improve oxygen reduction reactions (ORRs). ORRs are the reduction half-reaction that occurs at the cathode, reducing oxygen to water or hydrogen peroxides [68]. ORR catalysts are often used to improve the slow and complex ORR kinetics that occur in traditional air cathodes, directly effecting electric energy production [69]. Current ORR catalysts are expensive and scarce, with some disrupting oxygen and ion transfer. Hydrogel-derived cathodes have been fabricated in multiple studies to improve ORR performance.

Li et al. synthesised a microalgae hydrogel whereby *Chlorella pyrenoidosa*, conductive polyacrylonitrile fibre, and agar gel were applied to the cathode (Figure 3C) [58]. A 33% increase in maximum power density was produced in comparison to the traditional Pt electrode. This cathode however was limited by carbonate precipitation after prolonged operations. A tofu gel produced from soybeans was mixed with nitrogen and iron co-doped carbon to produce an ORR electrocatalyst [70]. They showed a maximum power output increase of 30.62% in comparison to the Pt electrode. The highest power density MFC was synthesised by Yang et al., whereby a nitrogen and iron chitosan gel was applied to activated carbon support [20]. The MFC had a shorter running time in comparison to the tofu gel and was slightly more expensive to fabricate than the other hydrogels (see Table 2).

#### 3.2.3. Membrane, Separator, and Electrolyte Hydrogels

Nafion or non-fluorinated polymers are traditionally used as the proton exchange membrane of MFCs due to their high proton conductivity, thermal and mechanical stability, and durability in the hydrated state. The high cost of these non-fluorinated membranes has resulted in their replacement with cheaper alternatives such as polyvinyl alcohol (PVA) [71]. The hydrogel form of PVA (PVA-H) can undergo repeated cycles of freezing and thawing to create an elastomer membrane [72]. The membrane showed excellent proton exchange, and when combined with the cathode to form an electrode assembly, further improved electricity production [71].

To our knowledge, PVA-H has only been applied to air cathodes and the MFCs are used to remove or degrade toxins from water sources. Chang et al. created a tubular MFC with the PVA-H PEM to remove benzene from ground water, while Wu et al. created a waterfall MFC to remove organics from molasses wastewater (Figure 3B) [57,72]. A similar application to remove azo dye was also conducted in 2017 [73]. Insufficient water uptake and retention by PVA-H can limit the proton transference ability of the material. In order to solve this, Liu et al. incorporated a water-retaining clay into the PVA-H, improving the proton conductivity by 2.87 times compared to the PVA-H MFC [74].

Evaporation is a common limitation in many air cathodes, especially miniature MFCs, resulting in unstable power generation [75]. Hydrogel polymers often contain hydrophilic functional groups; therefore when water evaporates, the internal pressure is reduced allowing the substrate to be pulled into the hydrogel. This phenomenon has been exploited to maintain ion transport in a horizontal air cathode design (Figure 3D) [59], to improve the contact between the ion exchange membrane in an MEA air cathode (Figure 3E) [60], and to create an auto feeding MFC that is able to draw up substrates mimicking transpiration [76]. Hydrogel electrolytes can be used to improve cathode potential without disrupting ion exchange.

### 3.3. Hydrogel-Based MECs

Hydrogels have also been used in MEC research for microbial immobilisation. Lescano et al. aimed to improve the efficiency of bioelectrochemical systems by improving the bacteria loading capacity of electrodes [77]. Graphene electrodes have been increasingly used due to their conductivity, high mechanical strength, and stability. However, the 2D structure limits bacterial attachments. Lescano et al. immobilised the electrogenic bacteria, *Geobacter sulfurreducens*, in a graphene hydrogel. The graphene hydrogel electrode resulted in a three-time increase in current density in comparison to that of the bare graphene electrode. The SEM images of the hydrogel showed large pore structures with dense microbial growth on the electrodes.

Gandu et al. on the other hand utilised hydrogels to isolate exoelectrogens from non-exoelectrogens, reducing competition for nutrients and resources [78]. The *Geobacter sulfurreducens* bacteria were immobilised in an alginate and chitosan hydrogel and the current density, COD removal, and hydrogen production was measured. The MEC systems were treated with acetate and wastewater, one containing no additional bacteria and the other containing free-growing microbes. The study found that the non-immobilised system performed better when fed with acetate; however, this performance decreased significantly when the analytes were switched to wastewater. The opposite occurred for the hydrogel-immobilised MECs. Genetic analysis of the communities showed that the hydrogel was able to maintain the community composition of the exoelectrogens, and protect it from external competitors [26].

Modification to improve the microbe-electrode interaction has been conducted on both MECs and MFCs [79]. Electrode and microbial modifications, cell immobilisation, and electrode material selection have been extensively researched [80,81,82]. Although hydrogels are not commonly used within MEC research, other microbe immobilisation methods are used [83]. Dubrovin et al. for example looked at encapsulating the microbial anode in a microfiltration membrane and Rozenfield et al. utilised a dialysis bag for the encapsulation [78,84].

Aerogels are macroporous versions of hydrogels that are also used in microbial entrapment of MECs. They are prepared via the sol–gel process followed by supercritical drying to remove all the liquid of the gel and replace it with air [85]. The following gel is extremely lightweight, highly macroporous, has good thermal conductivity, large surface area, and low density [86]. Hou et al. created an aerogel anode from Molybdenum disulphide (MoS2) and nitrogen-doped graphene (NG) [87]. The hybrid MoS_2_/NG has excellent electrocatalytic activity as well as good conductivity. By changing the structure from 2D to 3D aerogel, the porosity and surface area of the gel increased, further improving electrocatalytic activity. Another paper experimented on carbon aerogels and found that the hydrogen production was five times higher than that of the carbon cloth control [88]. The large surface area and large pores can promote greater microbial attachment, extracellular electron transfer, and electrocatalytic activity and make it a popular material for MECs.

**Table 2 sensors-23-00641-t002:** Summary of hydrogels and aerogels used in MFCs and MECs.

Hydrogel Applications in MET Technologies									
MET Type	Hydro/ Aerogel	Gel Use	Hydrogel Material	Anode (A)/ Cathode (C) Material	MFC/MEC Type	Maximum Power/ Current Density	Running Time	Other Remarks	Ref.
MFC	Hydrogel	Anode	Alginate, PANI, Titanium Dioxide, Graphite	Woven graphite fibre ©	Dual Chamber	7.88 W/m^3^	7 days ^1^	An increase in conductive materials results in the degradation of the gel matrix	[53]
	Hydrogel	Anode	BC, PANI	Graphite Pla© (C)	H-Type Dual Chamber	117 mW/m^2^	-	The bare graphite plate had a 1 mW/m^2^ density showing the significant effect of the hydrogel	[51]
	Hydrogel	Anode	BC, PANI, Titanium Dioxide, *Shewanella xiamenensis,* Ammonium Persulphate	Graphite S©ts (C)	Dual Chamber	38.89 W/m^3^	30 h ^1^	The maximum power density was 15-fold higher than bare BC electrodes	[52]
	Hydrogel	Anode	Sodium Alginate, PANI, CB	Grap©e Rod (C)	Dual Chamber	515 mW/m^2^	-	The modified anode had a power density of 1.38-fold higher than the bare anode	[54]
	Hydrogel	Anode	PPy, CMC, Titanium Nitride, CB©rbon Rod (C)	Cylindrical	Dual Chamber	14.11 W m^−3^	201 h ^2^	The modified anode had a power density of 4.72-fold higher than the bare anode	[55]
	Hydrogel	Anode	PPy, PANI, CNT, Iron Ox©	Carbon Rod (C)	Double Chamber	5901.49 mW/m^3^	-	The modified electrodes had a power output that was 1.33, 2.15, and 2.06 times greater than the PANI-PPy, PANI, and PPy anodes, respectively.	[56]
	Hydrogel	Anode	CNT, Chitosan, Carbon paper	Carbon cloth (C)	Single Chamber	132 mW/m^2^	-	Compared to the control, the maximum current and power density of the modified MFC increased by 23% and 65%, respectively.	[24]
	Hydrogel	Anode	GO, CNT, PNIPAM	Bare carbon cloth (A) (C)	Double Chamber	434 mW m^−2^	300 h ^1^	-	[62]
	Hydrogel	Anode	PPy, CNT	Carbon Paper (A) (C)	H-Type Dual Chamber	228 mW m^−2^		-	[61]
	Hydrogel	Anode	PPy, CMC, Nitrogen-doped Carbon Sponge	Graphite Rod (C)	Dual Chamber	4.88 W m^−3^,	153 h ^2^	The maximum power density of the modified electrodes were 1.34 and 1.71 times greater than the PPy/N-CNT/S and N-CNT/S bioanodes	[63]
	Hydrogel	Cathode	*Chlorella pyrenoidosa,* Polyacrylonitrile fibre, Agar	Carbon Fibre brush (A)		106.3 ± 5.9 mW m^−2^	-	Carbonate precipitate was found after a prolonged operation which is a limitation. Cost $0.35	[58]
	Hydrogel	Cathode	Nitrogen and FeCl_3_·6H_2_O self-doped activated carbon, Tofu gel synthesised from soybeans	Graphite felt (A)	H-Type Dual Chamber	863.40 ± 13.19 mW m^−2^	~400 h ^1^	-	[70]
	Hydrogel	Cathode	Activated Carbon, Fe(III)-chitosan-Nitrogen	Carbon brush (A)	Single chamber	2.4 ± 0.1 W m^−2^	120 h ^1^	Slightly more costly to fabricate, costing $2.2 m^−2^ of catalyst	[20]
	Hydrogel	Separator	PVA, Water	PVA-Coke (A) Carbon Cloth (C)	Tubular	38 mW m^−2^	120 days ^1^	Removed 95% of benzene Cost $0.0048 g^−1^ benzene	[57]
	Hydrogel	Separator	PVA, Carbon Cloth, Water	Carbon felt (A) Carbon Cloth (C)	Tubular	16.1 mW/m^2^	-	COD removal of 95.6%. Cost $25/m^2^	[72]
	Hydrogel	Separator	PVA, Water, Clay aggregate	Carbon Rod + Coke (A) Carbon Cloth (C)	Tubular	25.14 mW/m^2^	-	50 mL of Toluene was completely degraded in 6 days	[74]
	Hydrogel	Electrolyte	ECG gel	Activate Carbon (A) Carbon Cloth (C)	Tubular	6.1 W m^−3^	>6 months ^1^	-	[60]
	Hydrogel	Electrolyte	Oxidised Konjac Glucomannan/2-hydroxypropytrimethyl ammonium chloride chitosan	Carbon Brush (A) Activate Carbon (C)	Cubic reactor	1.0 ± 0.04 W/m^2^	-	Creation of an air cathode. Cost $0.12/m^2^	[59]
	Hydrogel	Electrolyte	Sodium Polyacrylate, Phosphate Buffer Solution	Stainless Steel Mesh coated in Carbon Black (A) Activated Carbon (C)	Cubic Reactor	295.5 W m^−3^	-	The system is inspired by the transpiration of plants	[76]
MEC	Hydrogel	Anode	GO, Ascorbic Acid, Carbon Cloth, Stainless Steel	Platinum wire (Working electrode) Ag/AgCl (Reference Electrode)	Single Chamber	-	-	-	[77]
	Hydrogel	Electrolyte	Alginate, Chitosan, Carbon Cloth	Carbon Cloth and Platinum (C)	Single Chamber	11.52 A m^−2^ At 0.2 V	-	COD removal of 78%	[26]
	Aerogel	Cathode	CB	Carbon Cloth loaded with different catalysts	Dual Chamber	0.36 mA cm^2^ at 0.8 V	-	Hydrogel production rate of 0.19 m^3^ H2 m^−3^ d^−1^	[87]
	Aerogel	Anode	Carbon Aerogel	Platinum	Dual Chamber	-	-	Hydrogel production (0.37 μmol cm^−2^ h^−1^) was 5 times higher than MECs containing a bio-carbon fibre anode with an external 0.3 V supply	[88]

^1^ Overall running time, ^2^ Running time of each cycle, “-” is equivalent to NA

## 4. Sensor Development Using METs and Hydrogel

In recent years, there has been growing interest in developing METs and hydrogel METs for sensing platforms, with MFC and MEC technologies being most commonly applied as sensors [89]. METs contain unique exoelectrogenic microbes that can act as bioreceptors, detecting a wide range of molecules [90]. These exoelectrogenic microbes are able to produce extracellular electrons, which can generate a current when transported from the anode to the cathode [91]. Electrical signals can be immediately read by a data processor, reducing the need for a transducer [1]. The current produced is highly dependent on the ability of the cell to oxidise organic molecules and the electron transfer rate and can therefore be used to determine analyte concentration [92].

### 4.1. MFC-Based Sensors

MFC-based sensors have been used across multiple industries including the medical, environmental, and robotics shown in Figure 1A. The main applications, however, lie in the environmental sensors used in water and wastewater treatment [93,94].

#### 4.1.1. Environmental Sensors for Water and Wastewater

##### Organic Matter Sensors

MFCs naturally breakdown organic matter and are able to convert chemical energy into electrical energy [95]. Previous research has determined that the current output of the MFC can be correlated with the concentration of organic matter and the charge output is correlated with the total amount of organic matter [96]. The changes in current output could therefore be correlated to the changes in the concentration of organic matter present in the water sample.

##### Biological Oxygen Demand (BOD) Sensor

The first MFC-based BOD sensor was reported in 1977, where *Clostridium butyricum* was able to estimate BOD levels in industrial wastewater [97]. Following this study, Kim et al. carried out a 5-year experiment, demonstrating stable BOD detection by the MFC and confirming the proportional relationship between columbic efficiency and organic matter in wastewater [98]. Electron acceptor diffusion into the anode chamber however can affect MFC performance as the electron acceptors can compete with the anodic biofilm for electrons, affecting microbial metabolism and reducing the columbic yield [99]. Respiratory inhibitors azide and cyanide have been shown to reduce the inhibitory effect of the electron acceptors, improving the sensitivity of the sensor [100]. Although Chang et al. showed that the respiratory inhibitors did not affect the sensor signal in the absence of electron acceptors, high concentrations of azide have been shown to change the microbial population, affecting the electron transport pathway [101]. A more recent paper used a low oxygen permeable proton exchange membrane to solve the oxygen diffusion problem. The resulting sensor showed a 62.5% improvement in detection range in comparison to a nafion MFC [102].

Multiple studies have looked into improving the sensitivity, selectivity, and detection range of sensors through modification of the electrodes and MFC configurations [103]. Pham et al. coated a graphite cathode in platinum to improve oxygen affinity [7]. The resulting sensor was able to detect low concentrations of BOD. The replacement of platinum with the highly catalytic manganese oxide showed an improvement in the detection limits of BOD and shorter response times [104,105]. Microbial attachment to the anode can also affect BOD detection. The covalent bonding of microbes to the anode was accomplished by electrodepositing carboxyl graphene and gold nanoparticles onto the anode. The improved microbial–electrode interaction reduced the detection time to 3 min [106]. Recent literature has shifted the focus to the creation of real-time monitoring autonomous sensors [99,107]. Pasternak et al. produced a sensor that would sound an alarm when the pollutant level exceeded a certain level and Tardy et al. automated the cleaning and washing steps of an MFC [108,109]. This can reduce the amount of manual sampling and monitoring needed.

##### Chemical Oxygen Demand (COD) Sensor

Most MFC-based COD sensors in literature are applied to monitor the COD levels in constructed wetlands (CW). CWs are a cost-effective method of wastewater treatment and COD levels are a good indicator of the organic removal rate by the CW [110]. Xu et al. successfully integrated an MFC sensor into CW to measure COD (acetate) concentrations [111]. Linear correlation between the voltage output and COD concentration was shown between 0–500 mg/L and 500–1000 mg/L. This sensor, however, did not account for other COD types or potential interference molecules and cannot be applied to more complex wastewater sources [112]. A similar paper tested the use of gravel or graphite-based anodes in COD assessments of domestic wastewater in CW [113]. Both materials were deemed suitable for domestic wastewater treatment, however, the detection time exceeded 10 h. Faster and more accurate detection of CODs is still required before it can be applied to real-world situations.

##### Volatile Fatty Acid (VFA) Sensor

VFAs are a common method to monitor the progress of the highly unstable anaerobic digestion (AD) process. High concentrations of VFAs can disrupt the anaerobic digestion (AD) process making VFAs an ideal analyte for detection [114]. Kuar et al. first described the ability of the MFC to quantify MFCs back in 2013 [114]. The MFC was able to efficiently breakdown and distinguish between three different types of VFAs. They were able to show proof of concept; however, the detection range was too high for real-world applications [115].

Single, double, and three-chambered MFCs have been designed for VFA production shown in Figure 4. Single-chambered MFCs are commonly used as shock sensors. Schievano et al. demonstrated that a VFA concentration of above 4000 mg/L will result in an inhibition of the system (Figure 4A) [116]. Double-chambered MFCs have been used to distinguish between VFAs and other organics in complex organic mixtures (Figure 4B) [117]. The study fed the inoculation media through the cathode, allowing for acetate to diffuse into the anode to be digested by the biofilm, reducing interference from other VFAs. Although complex organic molecules such as glucose had little effect, other VFAs did produce a current increase. Three-chambered MFCs as shown in Figure 4C, were designed to improve stability; however, they are highly costly [118]. Air cathodes have the same benefits as the three-chambered MFCs but have reduced operational costs [119]. Other AD effluents such as ammonium and fumarate have been demonstrated to influence the voltage output of MFC sensors [120]. Further exploration into improving the sensitivity and selectivity of the sensor towards VFA is necessary.

##### Nutrient Sensor

Nitrites and nitrates are present within the environment whether through natural causes such as the nitrogen cycle, or artificially through sewage or fertiliser leaching [122]. High levels of either type of ion are toxic to humans, animals, and plants, causing a variety of health effects [123]. Detection of these inorganic ions in the environment is therefore highly important. Nitrite ions affect the survivability of the microbes, resulting in a drop in MFC voltage when present [124]. MFCs can therefore be used as an early warning device. Wang et al. tested the effect of an open external circuit MFC sensor for the continual monitoring of nitrate ions [8]. The paper identified good sensitivity between nitrate concentrations of 10–40 mg/L and the sensor showed stable performance.

##### Toxicant Sensors

MFC can be developed as a toxicant sensor based on the inhibition effect of the toxicant on anode biofilm and current decrease. The main types of MFC-based toxicant sensors include heavy-metal biosensors and organic matter biosensors [99]. Few papers on acidic [125] and gaseous [126] toxicants have also been identified.

##### Heavy Metal Biosensors

Heavy metals are pollutants that are highly toxic even at low concentrations, making them a public health concern [127,128,129]. Heavy metal monitoring using MFCs has increased in recent years due to the complicated and time-consuming process of traditional monitoring systems [130]. Although some heavy metals are structural parts of biological materials (manganese, copper, and iron) or part of biological pathways (nickel, zinc, and magnesium), they are still harmful past a certain limit [130]. MFCs have been developed for the detection of a variety of heavy metals including Pb^2+^ [9], Fe^2+^ and Mn^2+^ [131], Ni^2+^ [132], Cd^2+^ [133], etc.

MFCs are commonly used as shock sensors due to the toxic effect that heavy metals can have on anodic processes. An early detection shock sensor was able to stably measure Cr^3+^ concentrations between 0.0125 and 5 mg/L [134]. Shock sensors are often single or short-term use due to the inhibition effect the toxicant has on the electrogenic activity of the microbes. The use of the cathode as the sensing element is in solution. Copper ions are able to cause a reduction in cathodic charge resistance, resulting in a decrease in voltage output [135]. A fast responding (20-s) copper sensor was developed based on MFC technology [10]. Xu et al. demonstrated that repeat exposure of the microbes to the target toxin can improve the robustness and tolerance of the microbes [136]. This shows the possibility of attaining and long-term shock sensor.

The ability of sensors to detect multiple heavy metals and differentiate between the different metals is important [137,138,139]. Lui et al. looked at the effect of Cr^6+^, Fe^3+^, NO^3−^, and acetate on the voltage output [140]. The paper found that 8 mg of Cr^6+^ caused inhibition whereas Fe^3+^ required 48 mg/L before a sharp voltage drop is seen. NO^3−^ has a slow voltage decrease whereas acetate creates a voltage increase. The paper demonstrates that different toxins have different voltage output profiles which could be used to differentiate between different toxins present in water samples

##### Toxic Organic Pollutant Biosensors

Organic toxicants are pollutants that are often sources of industrial waste which end up within the water system [141]. When present at high levels, these toxicants can result in major health problems [142]. MFC biosensors have been produced to detect a wide variety of organic pollutants, including phenolic compounds, such as Bisphenol A [143] and *P*-Nitrophenol (PNP) [144], formaldehyde [145,146], pesticides [132], and antibiotics [147].

Atrazine is one of the most commonly applied pesticides worldwide [148]. It targets the endocrine systems; therefore, high concentrations can be harmful to human health [149]. Environmental and operation conditions can often affect the performance of MFCs. Chouler and Lorenzo defined the optimal operational conditions for a miniature MFC biosensor for the detection of atrazine [148]. From their study, the pH had the greatest impact on sensor sensitivity, whereas temperature and ionic strength of the influent had minimal impact. Further optimisation was conducted in a later paper in which the effect of the cathodic materials on an MFC sensor was tested [150]. Out of carbon felt and indium tine oxide, the graphite felt cathode showed a faster response time and sensitivity to the EU legal limit of atrazine concentration in water, 0.1 μgL^−1^. The studies demonstrated that the determination of the optimal MFC parameters is important for good sensor performance.

Low levels of antibiotics can give rise to antibiotic resistance in microbial species, potentially leading to an outbreak. This is highly dangerous, requiring the need for real-time detection and removal of antibiotics from water systems [99]. Catal et al. were able to utilise MFCs to simultaneously detect and break down a glycosidic antibiotic (neomycin sulfate) [151]. The sensor was able to detect between 20–100 mgL^−1^ of the antibiotic. Another study created a panel of miniature MFC sensors to test for the effect of 10 beta-lactam antibiotics on two bacterial species [152]. The antibiotic effects and potential antibiotic resistance in electrogenic microbial species can be tested. The sensor had a 2–4 h response rate, faster than current methods of testing antibiotic resistance. This is highly beneficial in the medical field.

#### 4.1.2. Medical Sensors

##### Clinical Diagnostics

Within clinical diagnostics, MFC sensors can be used to detect different chemicals within different types of body fluids [153]. DNA contains all the information coding for the development and functionality of living organisms, therefore single mutations can have detrimental effects [154]. Mutations in the p53 tumour suppressor can result in the development of breast cancer. Asghary et al. developed a self-powered, label-free DNA sensor from a 2-chambered MFC that was able to discriminate between complimentary sequences by observing the difference in electrochemical output [155]. Their results showed high discrimination when spiked human serum samples were tested. Another common target molecule of biomedical analysis is glucose [156]. Li et al. created a reusable MFC-based sensor for selective glucose monitoring in urine [157]. The novel cylindrical sensor design was able to accurately detect a change in glucose concentration between 0.3–1.5 mmol. Other urine analytes were shown to have a negligible effect on the output voltage showing good selectivity to glucose, and the results were comparable to that of a commercial glucose sensor.

##### Infectious Species

The potential of MFCs to detect pathogens has been demonstrated in literature [158,159]. Hassan et al. were able to create a nanomaterials-based microbial sensor to determine the viability of the infectious *Streptomyces* spp. [158]. The paper identified that metabolically active and inactive bacterial cells resulted in different electrochemical signals. The sensor could also be used to determine the effects of different antibiotics on the species *E. coli* is an indicator of faecal contamination and therefore water contamination. Kim and Han created an MFC sensor to detect specific enzymes expressed in *E. coli* [159]. Their study verified the ability of the sensor accurately to quantify *E. coli* concentrations from different environmental samples.

#### 4.1.3. Other Sensing Applications

Outside of the environmental and medical field, MFC sensors have also been applied to detect other parameters. MFCs have successfully monitored metabolism as a signature of extra-terrestrial life [160]. Abreveya et al. published two papers back in 2010 which utilised MFCs to detect for photosynthetic and microbial metabolisms of different microorganisms as proof of extra-terrestrial life [161]. Figueredo et al. used the light and dark cycle to identify photosynthetic organisms [162]. Both studies demonstrated that greater power densities were obtained in samples containing metabolically active microorganisms. In addition to detect metabolism, Greenman et al. created a bio-bot from MFC technology to mimic the thermoreceptors of mammalian cells. The sensor developed is able to detect thermal stimulus of 1 °C and steer away and towards the thermal source, and continual thermal measurements could be taken at an interval of 160 s without the need for thermoreceptor recovery [163].

### 4.2. MEC-Based Sensors

Recent research has proposed the use of MECs for biosensor applications due to their ability to overcome thermodynamic limitations and catalyse bioelectrochemical reactions [164]. While this sector of research is fairly novel, MEC applications to environmental monitoring have been demonstrated. Examples are shown in Figure 5.

MFCs have commonly been employed to detect organic matter [165]. However, they are limited by the internal resistance and ORR at the cathode [90]. MEC can reduce internal resistance through the application of an external voltage overcoming some of the limitations of MFCs. Modin et al. developed an MEC-based BOD sensor that was able to detect BOD concentrations ranging from 32 to 1280 mg/L in 20 h [166]. To optimise the sensitivity and detection time of the BOD sensor, Wang et al. tested the effect of anode materials, inoculum source, and sensor configuration on sensor performance (Figure 5C) [167]. The study found that stacked MEC effluent showed the fastest response time and current generation when used as the inoculum in comparison to MFC effluent and sediment leachate. Miniature MEC designs, carbon felt anode, and reduced electrode spacing showed the greatest improvement in the sensor response by detecting concentration ranges of 10–500 mg/L of BOD within 5 min. The study demonstrated the potential of MECs to act as early warning sensors. The high voltage application has been shown to drain the accumulated electrons, reset the electrogenic capacity of the bacteria, and improve MEC sensitivity. Yuan et al. establish that treating the MEC to a 1.8 V for 1 min followed by the application of 1.2 volts can reset the bioanode sensors and reduce the effect of hysteresis [168]. The method however requires high energy consumption which can be very costly. VFA detection in AD using a MEC-based biosensor was demonstrated (Figure 5B) [169]. A linear correlation between the current density and VFA concentrations was observed under changing parameters, which demonstrates the potential use of MECs for VFA monitoring.

**Figure 5 sensors-23-00641-f005:**
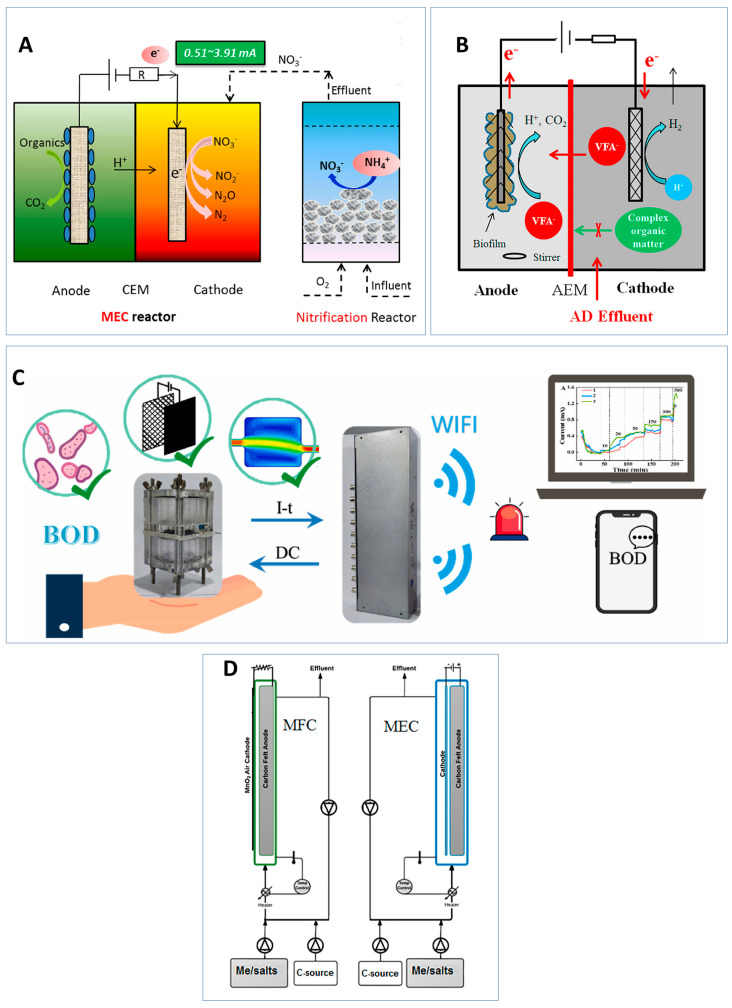
Examples of MEC-based sensors for environmental applications. (**A**) Nitrification–MEC loop system sensor for ammonia monitoring during wastewater treatment [170]; (**B**) the detection of VFAs in AD process using MEC sensor [169]; (**C**) optimising the MEC sensor performance by testing various anode material, inoculants, and sensor configuration [167]. (**D**) Schematic diagram of the MFC and MEC technology used to directly compare the sensing performance [92].

MECs have also been used to monitor inorganic and organic pollutants. Zhao et al. created an ammonium sensor that was connected to an external nitrification system shown in Figure 5A [170]. The study used a loop system whereby the effluent of a nitrification reactor is used to inoculate the cathode chamber of the MEC. The nitrification reactor converts ammonia to nitrate, and the nitrate is reduced as the electron acceptor. The study showed a linear relationship between the current and ammonia levels (0–62.1 mg NH_4_^+^-N/L), regardless of changes in the external power supply or wastewater pH. A miniature, three-electrode MEC was fabricated for the detection of three toxicants [171]. A decrease in current response was observed upon introduction of the toxicants and a detection range of 0.02–0.4 mM was observed for the toxicant imidazole. The sensor however is highly affected by the presence of oxygen due to the anaerobic nature of the microbes used. Trichloroacetic acid (TCA) is an organic toxicant produced during the disinfection process of wastewater treatments. It has high risks of carcinogenesis and is highly toxic to humans [172]. An MEC sensor containing nitrifying microorganisms was used for TCA detection. The study found that an increase in TCA concentration resulted in a decrease in nitrification but an increase in current output. The sensor was able to detect up to 5000 μg/L of TCA.

The advantages of using MECs as sensors have been demonstrated in literature; however, few studies directly compare the performance of MECs and MFCs. Adekunle et al. compared the ability of MFC and MEC to detect different concentrations of acetate, NH_4_^+^, Na^+^, Mg^2+^, and Fe^2+^ (Figure 5D) [92]. Three test phases were carried out in the experiments. Phase 1 spiked sensors with inorganic contaminants, while phase 2 continuously fed sensors with inorganic contaminants and Phase 3 switched the feeding stock with brewery wastewater. The MFC showed higher sensitivity to a larger range of toxicant compounds and had a shorter response time in comparison to MECs. Under continuous operations, linear correlations between steady-state current and sodium and ammonium sulphate concentrations were observed in both MFC and MEC sensors. A higher correlation between current and VFA concentration was shown in MFC sensors however, therefore only MFC tests were carried out for Phase 3 testing.

Although the performance of the MEC was inferior to that of the MFC, it has the potential to be used for biosensing applications. MEC technology is currently still in its infancy. Further research to determine the optimal conditions and design configurations can resolve the limitations currently faced by the technology [21]. Du et al. opened the door to coupling MECs to computing and the use of machine learning to improve the accuracy of toxicant detections and discrimination [173]. Additional development in this sector can bring MEC technology closer to being applied in real-world cases.

### 4.3. Hydrogel-Based MFC Sensor

To date, several studies on the application of hydrogels to MFC sensors have been published. These identified examples are shown in Figure 6.

The microbial biofilm plays a crucial role in the electricity production of MFCs. Previous research has aimed to enhance biofilm activity through genetic and microbial community engineering [176]. In the past, few pieces of literature have analysed individual microbial performance within MFCs and the ability to culture-specific microbial species [177]. Understanding the electrochemical activities of individual microbial species and being able to select high-performing microbes is important for enhancing MFC performance [174]. The screening of new electrogenic strains was demonstrated using an MFC array [178]. The study allowed a parallel comparison of the electrogenic performance of different microbial species in different environments. Compartmentalisation can reduce the effect that physical or environmental heterogeneities have on microbial activity, resulting in more accurate evaluations [179]. Mottet et al. encapsulated microbes in alginate–CNT gel beads to probe the microbial activity shown in Figure 6A [174]. The CNT can form conductive networks aiding electron transport and the alginate allows for nutrient exchange within the cell. The alginate–CNT gel showed good biocompatibility with the microbes and the technology has the potential to be applied to the detection and isolation of species in nature, the tracking of microbial metabolic activity as well as studies on species evolution and diversification under different environments.

Some authors have reported a potential sensor application of hydrogel MFCs. Winfield et al. produced a ceramic MFC to test the effect of oxygen on biofilm performance [175]. The MFC was designed so that the activated carbon and PVDF-coated carbon veil sheet covered the internal face of the ceramic cylinder, and the carbon veil anode was wrapped around the outside as shown in Figure 6B. To prevent biofouling and peeling of the cathode, a powdered hydrogel was introduced. As the electrochemically produced catholyte appears in the space, it causes the hydrogel to swell, improving contact between the cathode and ceramics. The study tested two different-sized ceramics and tested the effect of oxygen injections on the power output. A 5 mL injection of air resulted in a 42% decrease in power output in the small MFC, demonstrating its potential use as a dissolved oxygen (DO) sensor. The cathode is the sensing module in most MFC DO sensors. However, sensor signals are often influenced by external molecules such as organic matter, therefore an anode DO sensor might be a better alternative [180].

VFA concentrations are a good way of monitoring the performance of anaerobic and aerobic processors. An MFC-type biosensor was produced by Kuar et al. for the detection of the three most common types of VFAs, acetate, butyrate, and propionate [114]. The study demonstrated that discrimination between different VFA species can be shown by the current output. To enhance the temporal stability, reproducibility, and selective sensitivity of the sensor to different VFAs, three natural polymers and three PPys were used to modify carbon paper bioanodes shown in Table 3 [115]. The Ppy^+^ modified bioanode reduced the start-up time by 50 h and >200 h for acetate and propionate respectively and increased the maximum cell voltage for all three VFAs. The agarose and polyacrylamide gels showed an increase in voltage; however, the PVA gel showed a decrease in voltage due to mass transport limitations. The polymer gels also only remained stable for 3–4 days due to the lack of mechanical and chemical stability between the electrode and gel matrix. Overall, the Ppy^+^-modified bioanode demonstrated stable and reproducible results over 2 weeks. Recovery times for the Ppy^+^ modified bioanode were also significantly faster than the control showing increased microbial resistance. Although the sensing range can be improved, the study demonstrates the potential use of hydrogels to enhance sensor performance.

## 5. Outlook

MET is a fast-emerging sector that is increasingly being applied to the sensor industry. Its ability to bypass the use of an external transducer makes it highly popular for sensor applications. MFCs and MECs have both been developed for sensing applications due to their ability to detect a wide range of analytes. MFC has great potential for sensor applications, with many pieces of research demonstrating sustainable and long-term operations. Research on MEC-based sensors is limited in comparison to MFC research. Unlike MFCs, MECs are not autonomous and require energy input. This can affect the long-term stability and durability of MEC sensors. MEC-based sensor research is still novel. Further research on identification of the optimal conditions of MECs will be highly beneficial to advancing the field.

Poor sensitivity, selectivity, detection rates, and detection limits are major limitations that are consistently commented on throughout both MFC- and MEC-based sensor research. Hydrogels have the potential to improve the sensitivity and selectivity of existing MFC sensors due to the range of unique properties they possess. The hydrogel industry is a fast-growing sector that has gained a significant amount of scientific interest in recent years. It has been extensively applied to various sensing platforms, exhibiting improved sensor performance.

Hydrogel application to METs is gaining interest; however, publications are still limited. Within the MFC industry, research has almost exclusively focused on tuning hydrogel properties to improve the electricity output of MFC efficiency with a significant portion of the research focusing on anode hydrogels. The benefits of using hydrogels for MFC applications are clear, and it is expected that more research focusing on the effects of hydrogel composites on MFC performances will appear. Tailoring hydrogel composites for different MFC components should be further investigated. To date, most MET-based sensors are hand-made, with a time-consuming manufacturing process and low accuracy. Hydrogels have high tailorability and flexibility, which are ideal materials for automated manufacturing, such as 3D printing [181]. Evidently, this unique advantage will substantially benefit the practical application of METs sensors in future. Additionally, natural polymer-based hydrogels are environmentally friendly, which can avoid potential plastic pollution [182,183,184,185].

Although the benefits of hydrogels application to METs and sensors are noted in the literature, research on hydrogel-based MET sensors is fractional. The lack of publications reflects the novelty of this sector of research, highlighting a potential research gap. The few existing studies on hydrogel-based MET sensors used MFC and simplistic, non-composite hydrogels were used. These studies, however, have unlocked a new, innovative area of research that could significantly advance and broaden the applications of sensor research.

Future research should first focus on developing the use of MECs as sensing platforms, and concentration on optimising different materials and design configurations. A large variation of hydrogel composites exists, each with tailorable properties. The effect of these hydrogel composites on MET applications should also be explored. It is clear that METs show promising potential as sensing platforms. Future research should expand existing research by investigating the effect of hydrogel on the performance of MET-derived sensors. This could result in new developments within both the sensor and environmental technological fields.

## Figures and Tables

**Figure 1 sensors-23-00641-f001:**
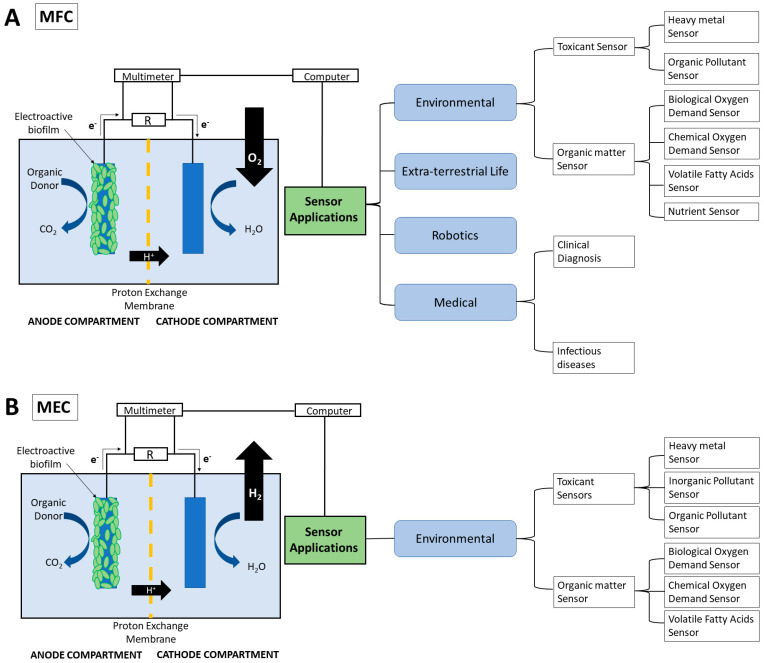
Schematic representation of the (**A**) microbial fuel cell and (**B**) microbial electrolysis cells and the anodic and cathodic reactions that occur. The sensor application of each MET type is also summarised.

**Figure 2 sensors-23-00641-f002:**
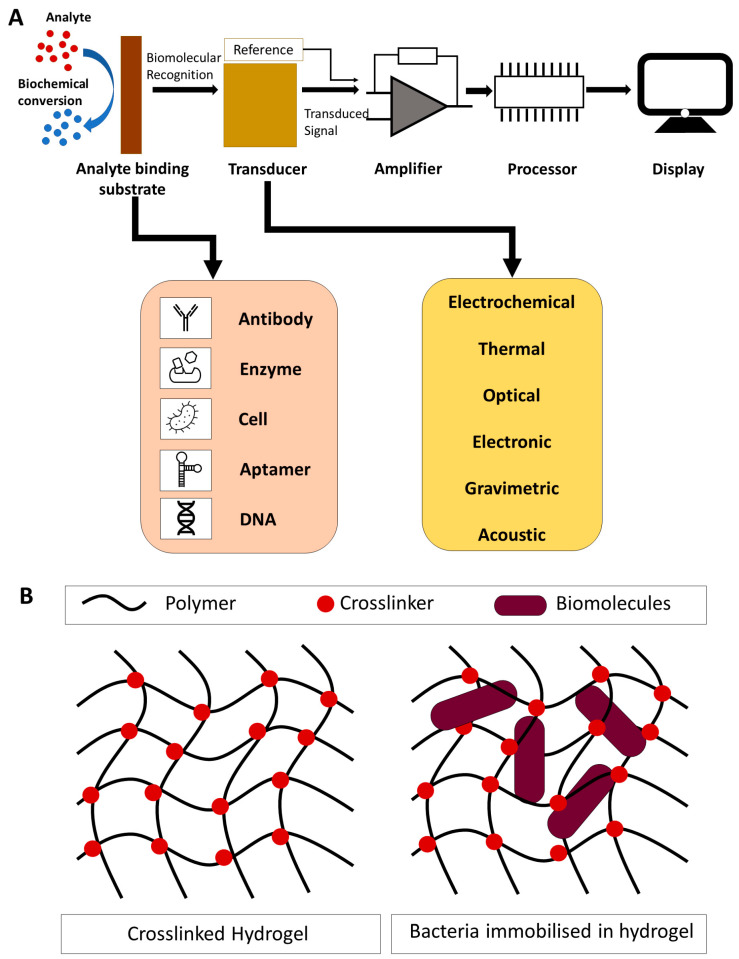
Schematic diagram to show the biosensor system structure. (**A**) The different bioreceptor and transducer categories are characterised; (**B**) the general hydrogel structure and its ability to immobilise microbes.

**Figure 4 sensors-23-00641-f004:**
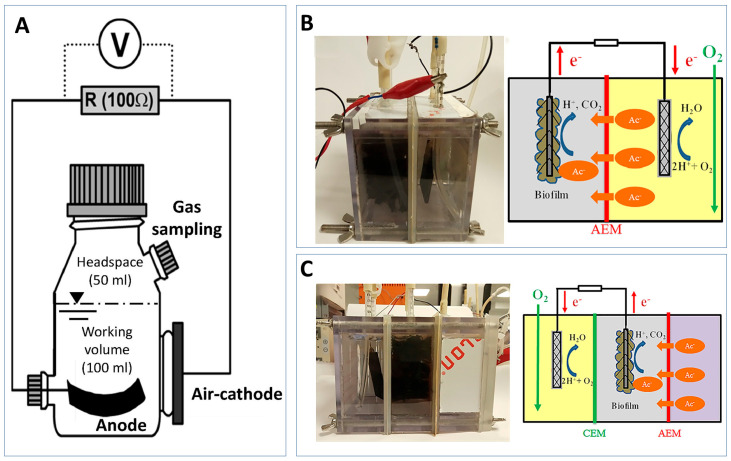
Summary of (**A**) Single-chambered [116], (**B**) double-chambered [117,121], and (**C**) three-chambered [118] MFCs for VFA detection.

**Figure 6 sensors-23-00641-f006:**
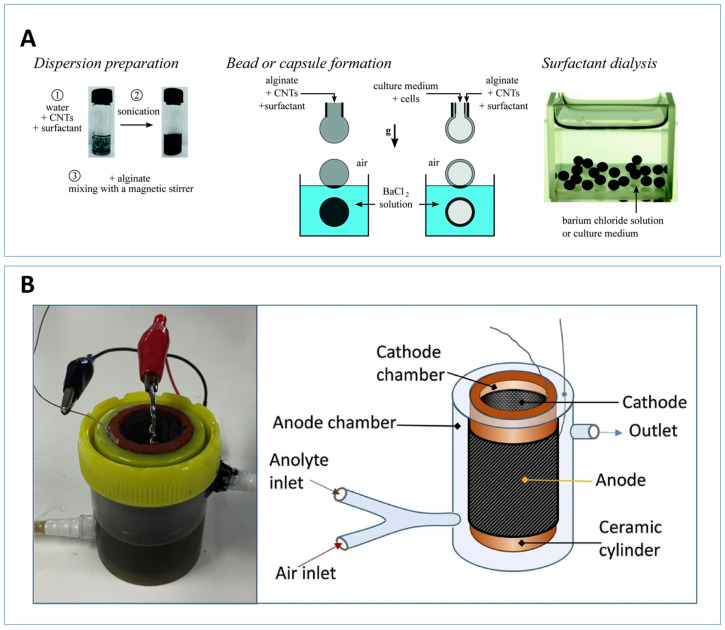
Examples of hydrogel based MFCs used in sensor applications. (**A**) Encapsulation of microbes in alginate/CNT hydrogel beads for the probing of microbial activity [174]; (**B**) ceramic MFC containing a catholyte expanding hydrogel, improving the contact between the cathode and the ceramic [175].

**Table 1 sensors-23-00641-t001:** Advanced keyword and search terms applied to the Web of Science database and the number of publications identified.

Key Term	Number of Publications
“Sensor”	791,393
“Sensor” and “Hydrogel”	4791
“Sensor” and “Microbial Fuel Cell”	591
“Microbial Fuel Cell” and “Hydrogel”	80

**Table 3 sensors-23-00641-t003:** Different anode modifications made for VFA sensing [115].

Polymer Modified Bioanode	Pyrrole Modified Bioanode
Agarose and mediator (AG + NR)	pyrrole (Ppy)
Polyacrylamide and mediator (PAM + NR)	pyrrole propanoic acid (Ppy^−^)
Polyvinyl alcohol (PVA) + Calcium Alginate + Carbon Powder	pyrrole alkylammonium (Ppy^+^)

## Data Availability

All the raw data are available by contacting L.X.

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
