# Peer review of "Recent Implementations of Hydrogel-Based Microbial Electrochemical Technologies (METs) in Sensing Applications"

_sensors, 2023, doi:10.3390/s23020641_

Round 1

Reviewer 1 Report

Dear authors

This is a well-written review related to hydrogel applications inmicrobial emetrochemcial technologies and sensors. It can be accepted for publication after a major revision with following issues.

1. The title of this review is not appropriate and it does not correctly presents the contents of this review. I suggest it should be revised as "Recent implementations of hydrogel-based microbial electrochamical technologies (METs) in sensing applications"

2. Section 2 is not necessary. Many published reviews have summaried and discussed the classification of hydrogels. Moreover, this section simply calssified in two types of hydrogels, i.e. synthetic and natural hydrogels, which is very common and well-known.

3. Instead, section 2 should provide a deep discussion about MFCs and MECs related to their difinition, differences, properties, advatages.

4. Table 1 shoud be presented as a scheme

5. I think this review should focus on the sensor applications. Other applications should be removed (4.1.3)

Author Response

Many thanks for taking the time to review the paper.  please see the attached response and revision.

Reviewer 2 Report

I noticed some figures were not presented well, such as Figure 4, and 5.

Author Response

Point 1: I noticed some figures were not presented well, such as Figure 4, and 5.

Response 1: Many thanks for taking the time to review the paper. As suggested, figure 4 and 5 have been reconfigured to have more space between each figure. Each figure has been downloaded with the highest resolution to ensure clarity. Figure 3 and 6 have also been changed to match this style

Please see the revision in figs 3, 4, 5, 6.

Reviewer 3 Report

The review manuscript describes the hydrogel-based microbial electrochemical technologies, in particular microbial fuel and electroanalysis cells. It covers wide variety of hydrogel types used in METs and thus provides a nice summary, including their applications. I think that the review is of good scientific quality and can be published in Sensors journal. 

Author Response

Point 1: The review manuscript describes the hydrogel-based microbial electrochemical technologies, in particular microbial fuel and electroanalysis cells. It covers wide variety of hydrogel types used in METs and thus provides a nice summary, including their applications. I think that the review is of good scientific quality and can be published in Sensors journal.

Response 1: Thank you for taking the time for the review and the comments.

Round 2

Reviewer 1 Report

It can be accepted for publication